

# Innate immunity, therapeutic targets and monoclonal antibodies in SARS-CoV-2 infection

Mubashir Nazir[1], Ishfaq Rashid Mir[2], Shabir Ahmad Lone[1], Ghazala Muteeb[3], Ragib Alam[4], Anis Bashir Fomda[5], Nida Khan[6], Asim Azhar[7], Bashir Ahmad Fomda[1] and Wajihul Hasan Khan[4]

[1] Department of Microbiology, Sher-I-Kashmir Institute of Medical Sciences Soura, Srinagar, Jammu and Kashmir, India
[2] Department of Immunology and Molecular Medicine, Sher-I-Kashmir Institute of Medical Sciences Soura, Srinagar, Jammu and Kashmir, India
[3] Department of Nursing, College of Applied Medical Science, King Faisal University, Al-Ahsa, Saudi Arabia
[4] Department of Microbiology, All India Institute of Medical Sciences, New Delhi, Delhi, India
[5] Department of Psychiatry, All India Institute of Medical Sciences, New Delhi, Delhi, India
[6] Department of Chemical Engineering, Indian Institute of Technology, Delhi, New Delhi, India
[7] Interdisciplinary Biotechnology Unit, Aligarh Muslim University, Aligarh, Uttar Pradesh, India

Corresponding authors
Ghazala Muteeb, graza@kfu.edu.sa
Wajihul Hasan Khan, wajihulbiotech@gmail.com

## ABSTRACT

COVID-19 (coronavirus disease 2019), caused by SARS-CoV-2 (severe acute respiratory syndrome coronavirus 2), stands as one of the most severe pandemics the world has ever faced in recent times. SARS-CoV-2 infection exhibits a wide range of symptoms, varying from severe manifestations to mild cases and even asymptomatic carriers. This diversity stems from a multitude of factors, including genetic predisposition, viral variants, and immune status. During SARS-CoV-2 infection, the immune system engages pattern recognition receptors, setting off a series of intricate signalling cascades. These cascades culminate in the activation of innate immune responses, including induction of type I and type III interferons. The emerging variants of SARS-CoV-2 pose challenges to the innate immune system defense. Therefore, investigating the innate immune response is crucial for effectively combating SARS-CoV-2 and its variants. The cyclic guanosine monophosphate-adenosine monophoshate synthase-stimulator of interferon genes (cGAS-STING) pathway, a critical innate immune mechanism, represents a promising target for intervention at multiple stages to reduce the severity and progression of SARS-CoV-2 infection. This review explores innate immunity in SARS-CoV-2 infection and other immune responses critical for SARS-CoV-2 defence. As part of the therapeutic approach, we extend our review to highlight monoclonal antibodies (mAbs) as emerging and effective therapeutics for controlling SARS-CoV-2 by targeting different stages of the innate immune system. A diverse range of mAbs has been explored to address specific targets within the innate immune pathways. A deep understanding of innate immunity and targeted monoclonal therapeutics will be instrumental in combating viruses and their variants, laying the foundation for enhanced treatment and therapeutic strategies.

## INTRODUCTION

SARS-CoV-2 is a member of the a/ß-Corona family, characterized by its enclosed, spherical structure. It possesses a non-segmented, positive single-stranded RNA (ssRNA) genome of approximately 30 kilobase pairs (kbp) shielded by the helical capsid made by the nucleocapsid (N) protein and enclosed by an envelope protein (E). The structure of the SARS-CoV-2 protein contains envelope (E) and membrane (M) proteins that help in virus assembly, and spike (S) protein allows the virus to enter the hosts. Among these E, M, and S proteins, the S protein size is too large (180–200 kDa) and appears like a crown (*Lan et al., 2020*) (Fig. 1A). Among seven viruses (SARS-CoV-2, SARS-CoV, MERS-CoV, HCoV-229E, HCoV-NL63, HCoV-OC43, HCoV-HKU1) associated with typical respiratory infections, four (HCoV-229E, HCoV-NL63, HCoV-OC43, HCoV-HKU1) cause harmless seasonal infections, while the remaining three SARS-CoV, MERS-CoV, and the recently identified SARS-CoV-2 pose a higher risk of lethality. SARS-CoV emerged in China in November 2002 and infected nearly 8,100 individuals, with a mortality rate of 9.6%, resulting in 774 deaths. MERS-CoV, transmitted from camels to humans, appeared a decade later, spreading globally over six years, infecting 2,143 individuals, and resulting in 750 deaths with a mortality of 34.9% (*Marschalek, 2023*).

In 2019, the SARS-CoV-2 emerged and rapidly spread across the globe. The total confirmed cases of SARS-CoV-2 have surged to a staggering 777.35 million, underscoring the extensive reach of the virus (https://covid19.who.int/). Since the beginning of the COVID-19 pandemic, several new variants of concern have emerged, including Alpha (B.1.1.7), Beta (B.1.351), Gamma (P.1), Delta (B.1.617.2), and Omicron (B.1.1.529). These variants are associated with increased transmissibility and virulence. Currently, the World Health Organization (WHO) is tracking various SARS-CoV-2 variants, which include a variant of interest (VOI): JN.1, and variants under monitoring (VUMs): JN.1.18, KP.2, KP.3, KP.3.1.1, LB.1, and XEC. The COVID-19 pandemic and emerging SARS-CoV-2 variants highlight the immense challenges and emphasize the urgent need for ongoing global efforts to mitigate its impact and prevent further losses (*Zeyaullah et al., 2021*). Many promising strategies demonstrating significant effort in the fight against SARS-CoV-2 are being investigated although challenges remain (*Khan et al., 2024*; *Khan et al., 2021b*; *Khan et al., 2022*; *Sharma et al., 2023*; *Zeyaullah et al., 2023*; *Zhou et al., 2021*). However, SARS-CoV-2 has developed several ways to avoid or circumvent immune mechanisms, allowing it to infect and spread throughout the host successfully (*Foxman, 2024*; *Zaidi & Singh, 2024*).

The innate immune response plays a crucial role in defending against SARS-CoV-2. Its primary functions include limiting viral entry, blocking viral translation and replication, and preventing the release of new infectious virions. Additionally, it facilitates the identification and elimination of infected cells while accelerating the development of an adaptive immune response (*Hoffmann, Schneider & Rice, 2015*; *Sievers et al., 2024*). However, in severe COVID-19 cases, an overactive immune response can lead to excessive inflammation, significantly impacting disease progression. A key component of this inflammatory response is the interferon cascade, which plays a critical role in SARS-CoV-2

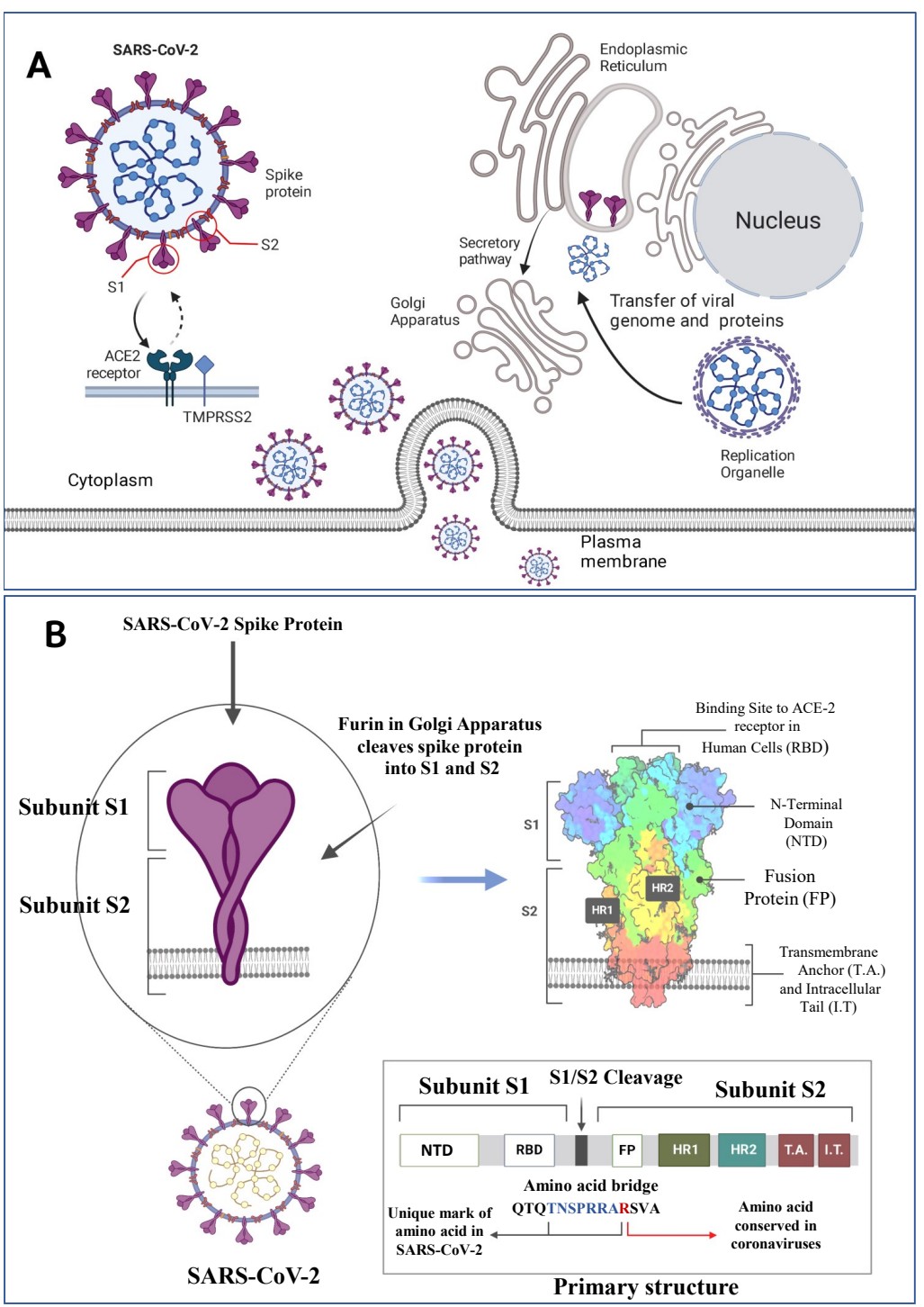

**Figure 1  SARS-CoV-2 structure, viral host interaction and virion formation in the host cells.** (A) The SARS-CoV-2 structure, and maturation involve the formation of new membranous structures known as "replication organelles" near the cell nucleus. These structures are surrounded by double membranes and are believed to originate from the endoplasmic reticulum (ER).

**Figure 1 (...continued)**
These organelles serve as a location for viral replication complexes, isolating them from the cells innate immune response. The process of virus assembly starts with the synthesis of viral proteins and genomic RNA at the replication site. These components are then transported, through an unknown mechanism, to the ER-Golgi intermediate compartment (ERGIC), where assembly and budding of new viruses occur. (B) The SARS-CoV-2 spike (S) protein is a crucial component of the virus responsible for binding to human cells and facilitating infection. It consists of various subdomains with distinct functions. One of the essential parts of the S protein is the receptor-binding domain (RBD), responsible for recognizing and binding to the human cell receptor ACE2 (Angiotensin-Converting Enzyme 2). The RBD has three distinct antigenic sites (AS-1, AS-2, and AS-3), which are regions that can trigger an immune response. RBD can be one of the targets for the monoclonal antibodies as depicted above (Figure created on biorender.com).

infection (*Merad & Martin, 2020*). In some individuals, cytokine release syndrome (CRS) occurs, characterized by an overwhelming cytokine surge that leads to acute respiratory distress syndrome (ARDS) and secondary hemophagocytic lymphohistiocytosis (sHLH) (*Moore & June, 2020*; *Quan et al., 2020*). To counterbalance this excessive inflammation, the innate immune system employs endogenous feedback mechanisms, utilizing cytokines such as IL-4, IL-10, IL-11, and IL-13 to promote an anti-inflammatory state.

The impact of COVID-19 varies significantly among patient populations, with immunocompromised individuals (cancer patients, organ transplant recipients, individuals with HIV, autoimmune diseases, or immunosuppressive therapy) facing a heightened risk. Among solid organ transplant recipients, lung transplant recipients exhibit the highest fatality rate (*Hall et al., 2022*). Immunocompromised individuals are at higher risk for severe COVID-19 complications, with mortality rates increasing by 5 to 6% compared to the normal population (*Liang et al., 2020*; *Wu & McGoogan, 2020*).

Understanding the innate immune response and the pathways that can trigger SARS-CoV-2 viral infections is crucial for precise and targeted therapeutic approaches. The type-I interferon (IFN-1) system plays a critical role in the innate immune response. Dysfunctional early IFN-1 signaling is a key feature of severe COVID-19 and is linked to reduced viral clearance (*Hadjadj et al., 2020*). A diminished or absent IFN-α response often precedes clinical worsening and the need for intensive care, marking the most severe or critical cases that require invasive ventilation. These cases show significantly lower expression of six interferon-stimulated genes (ISGs), which define the IFN-1 signature, compared to mild to moderate cases with elevated IFN-α levels (*Fraser et al., 2023*; *Zhang et al., 2021a*). During the replication and transcription of the coronavirus genome, double-stranded RNA (dsRNA) is produced (*Kindler, Thiel & Weber, 2016*), which can be identified by RIG-I-like receptors (RLRs) such as retinoic acid-inducible gene I (RIG-I) and/or melanoma differentiation-associated protein 5 (MDA5) in the cytoplasm(*Li, Liu & Zhang, 2010*; *Roth-Cross, Bender & Weiss, 2008*). Alternatively, toll-like receptors (TLRs) in endosomes can also recognize dsRNA (*Totura et al., 2015*). The two-caspase activation and recruitment domains (CARDs) of RIG-I and MDA5 can interact with the mitochondrial antiviral signaling protein (MAVS)(*Seth et al., 2005*). A study highlights the interaction between SARS-CoV-2 and the host's innate immune system, revealing how viral proteins play a role in evading immune defences (*Lei et al., 2020*). Patients with weakened immune systems, such as those with rheumatological

conditions or lung transplants, often experience severe COVID-19 symptoms and prolonged hospital stays (*Gianfrancesco et al., 2020*; *Heldman et al., 2021*). Individuals with hematologic malignancies, face a heightened risk of SARS-CoV-2-related morbidity and mortality. This is due to immune deficiencies that hinder the prevention, treatment, and elimination of the virus (*DeWolf et al., 2022*). Data from previous studies suggests that some medications often used to manage immune-mediated inflammatory conditions, such as cytokine inhibitors, could help reduce the severity of COVID-19 (*Fagni et al., 2021*; *Heimfarth et al., 2020*). However, treatments like glucocorticoids and those targeting B-cells may negatively impact COVID-19 outcomes (*Bruscoli et al., 2022*; *Fagni et al., 2021*). Treatment options include neutralizing mAbs such as regdanvimab, teleseminar, and imdevimab, which specifically target the spike protein of SARS-CoV-2 to inhibit viral replication and reduce disease severity (*Miguez-Rey et al., 2022*). Among the other factors that manage the progression and severity of SARS-CoV-2 is vitamin D. The BsmI b allele and bb genotype have been linked to an increased likelihood of hospitalization due to SARS-CoV-2 infection, potentially due to the association of the b allele with reduced vitamin D levels (*Aci et al., 2024*). The insertion/deletion polymorphism of the endothelial growth factor (VEGF) gene, specifically the DD genotype and D allele, has been associated with vitamin D levels in patients with COVID-19 (*Yigit et al., 2023*). To manage SARS-CoV-2 infection, immunomodulatory therapies, such as corticosteroids and monoclonal antibodies, play a crucial role, especially in severe cases where immune dysregulation drives disease progression. Corticosteroids (*e.g.*, dexamethasone, prednisolone) are particularly beneficial in cases involving hyperinflammation and cytokine storm, helping to reduce excessive immune activation and improve patient outcomes. As research continues, optimizing immunomodulatory interventions for different patient populations, especially those who are immuno-compromised, remains a critical area of focus.

The antibody-based interventions have paved the way for advancements in technology and production methods, ultimately creating more potent and promising therapeutic approaches in the form of mAbs as an important tool for pandemic preparedness. The target audience for this review will be scientists/clinicians/health workers and those interested to know more about the native immune response in SARS-CoV-2 infection and exploring potential treatments that target the viral proteins involved in these immune responses.

## SEARCH METHODOLOGY

In this review, we searched the literature on PubMed, Google Scholar, ScienceDirect, Google search engine, and Scopus by using the following terms COVID-19 immunity, SARS-CoV-2 immunity, innate immunity/COVID-19, innate immune response to SARS-CoV-2, SARS-CoV-2 protein/immune target, therapeutic strategy/SARS-CoV-2, monoclonal antibody in COVID-19 and COVID-19 therapeutic antibody. We also included reports from the Centers for Disease Control and Prevention (CDC), and the World Health Organization (WHO). The search terms were strategically broadened to ensure the inclusion of all relevant studies. Two authors independently screened the literature for reproducibility. Non-COVID-19 and non-English language studies were used as exclusion criteria.

# RECEPTOR-LIGAND INTERACTIONS IN SARS-COV-2 INFECTION

SARS-CoV-2 are detected through their pathogen-associated molecular patterns (PAMPS), when bound to host cell pattern recognition receptors (PRRs) and elicit innate immune responses against them (*Badia, Garcia-Vidal & Ballana, 2022*; *Takeuchi & Akira, 2010*). PAMPs in viruses primarily include viral nucleic acids, such as ssRNA, dsRNA, unmethylated CpG DNA and viral proteins, which are recognized by the host immune system to trigger an innate immune response. The interaction between PAMPs and PRRs activates the innate immune responses in the form of upregulation of complement proteins, secretion of antimicrobial agents, cytokine signalling, activation, and recruitment of phagocytic cells like, macrophages and natural killer (NK) cells (*Tosi & Immunology, 2005*).

The S protein of SARS-CoV-2 interacts with ACE2, a critical regulator of the renin-angiotensin system, as a cellular entry receptor for the invasion of virus into human cells. The S1 protein consists of two domains: an N-terminal domain (NTD) and a receptor-binding domain (RBD). The RBD interacts with the peptidase domain of ACE2 using a receptor-binding motif (RBM), while the exact function of the NTD remains unclear, it might be responsible for recognizing specific sugar structures during initial attachment. This recognition could aid the transition of the S protein from perfusion to post-fusion conformation. Antibodies binding to specific epitopes on the NTD have been proven to hinder SARS-CoV-2 infection (Fig. 1B) (*Jackson et al., 2022*). SARS-CoV-2 activates the innate immune responses through PAMPs (like viral RNAs and oxidized phospholipids) and the infection was restricted to entry gates of the human body (*Danladi & Sabir, 2021*; *Mihaescu et al., 2021*). Cells containing NLRP3 (NOD-like receptor family pyrin domain-containing 3) inflammasome like macrophages, epithelial cells and endothelial cells are activated by SARS-CoV-2 through a NOD-like receptor family, helping in caspase-1 activation. Caspase-1 activation leads to the cleavage of proinflammatory cytokine IL-1 into its physiologically active IL-1β and IL-18. Additionally, TLR-3, -7, -8, and -9 receptors respond to viral RNA and trigger the NF-κB pathway that activate the pro-inflammatory cytokine cascade (*Khan et al., 2021a*; *Lee, Channappanavar & Kanneganti, 2020*). When SARS-CoV-2 enters the body, it binds to the ACE2 receptor, triggering conformational changes in the S1 subunit (*Jackson et al., 2022*). This is followed by the cleavage of S2 by cellular proteases such as TMPRSS2 or cathepsin L (*Trougakos et al., 2021*). This chain of events is detected by various host pattern recognition receptors (PRRs) (*Diamond & Kanneganti, 2022*). The TLR family, RIG-I-like receptors (RLRs), and nucleotide-binding oligomerization domain (NOD)-like receptors (NLRs) are the three main families of PRRs (*Kanneganti, 2020*). Activation of these receptors through aberrant signalling pathways leads to over-activation of inflammatory cytokines and chemokines. The diversity among the PRRs in the form of extracellular receptors like TLR and dectins, which detect the extracellular PAMPs, and intracellular receptors including RIG-1-like receptors (RLRs), NOD-like receptors (NLRs), AIM2-like receptors (ALRs), which recognize the intracellular PAMPs (*Dawoodi, Rizvi & Zaidi, 2024*). PAMPs can simultaneously bind to multiple

receptors and other signalling pathways to drive extended cellular events including cell death (*Franz & Kagan, 2017*; *Kagan & Barton, 2015*). A deeper understanding of the pathophysiological mechanisms involved in innate immunity is crucial for developing effective preventive and therapeutic strategies against COVID-19. The main families of PRRs are summarized below.

## TLRs and SARS-CoV-2

TLRs serve as the primary defence mechanism for the innate immune system, protecting the host from pathogens. There are ten TLR family members, half of which are in the cell membrane, and the other half are found in endosomes. The X-chromosome harbours tandem duplicated genes known as TLR7 and TLR8. These genes are found on the membrane of the endosome and capable of identifying synthetic oligoribonucleotides such imidazoquinolinone, imiquimod, and R-848, as well as ssRNA. As a result, they had a part in the identification of the SARS-CoV-2 genome (*De Groot & Bontrop, 2020*). Whole-genome sequencing analyses of SARS-CoV, MERS-CoV, and SARS-CoV-2 have indicated a potentially heightened involvement of TLR7 in the pathogenic mechanisms of SARS-CoV-2, relative to SARS-CoV and MERS-CoV. This heightened involvement is suggested to arise from the greater abundance of single-stranded RNA motifs within SARS-CoV-2, facilitating increased binding affinity to TLR7 (*Van der Made et al., 2020*). The ssRNA virus, SARS-CoV-2, has been associated with at least six TLRs in viral recognition, namely TLR2, TLR3, TLR4, TLR7, TLR8, and TLR9. TLR2 and TLR4 can identify viral structural and non-structural proteins outside the cell (*Tyrkalska et al., 2023*). Human macrophages have been shown to engage TLR4 signalling in response to the SARS-CoV-2 spike protein S1 subunit, which causes pro-inflammatory reactions (*Shirato & Kizaki, 2021*). Inflammatory cytokines are produced by TLR2 upon sensing the SARS-CoV-2 envelope protein (*Zheng et al., 2021*). TLR3 recognizes double-stranded RNA during viral replication. Following macrophage engulfment of SARS-CoV-2, TLR7/TLR8 identifies the genomic RNAs released from the virions, activating subsequent signalling cascades. During pulmonary SARS-CoV-2 infection, TLR7 primarily regulates innate immunity by triggering NF-kB transduction and causing the release of pro-inflammatory cytokines (*Bortolotti et al., 2021*; *Planès et al., 2022*). However, research on TLR9's role in SARS-CoV-2 infection is still limited. When SARS-CoV-2 infects endothelium cells, mitochondrial dysfunction raises mitochondrial DNA (mtDNA) levels and activates TLR9 (*Costa et al., 2022*). TLR4 may play a major role in the pathophysiology of SARS-CoV-2 by causing abnormal inflammation (*Aboudounya & Heads, 2021*; *Sohn et al., 2020*). TLRs present promising targets for controlling infection during the early stages of the disease and for developing vaccines against SARS-CoV-2 (*Khanmohammadi & Rezaei, 2021*). Agonists targeting TLRs may induce a robust immune response in COVID-19 (*Florindo et al., 2020*).

## RLRs and SARS-CoV-2

RLRs comprise three related proteins viz. RIG-I, (or DDX58), MDA5, and laboratory of genetics and physiology 2 (LGP2). During SARS-CoV-2 infection, RIG-I and MDA5 primarily recognize and inhibit viral replication by identifying viral intermediate dsRNA.

Activated RLRs typically interact with mitochondrial antiviral signalling protein (MAVS) to regulate the IFN I and III pathways (*Yin et al., 2021*). The activity of downstream ISGs, including LY6E, AXIN2, CH25H, EPST1I, GBP5, IFIH1, IFITM2, and IFITM3, has been shown to inhibit SARS-CoV-2 replication and entry (*Martin-Sancho et al., 2021*). SARS-CoV-2 can interfere with the RLR signaling pathway in two ways: either through deubiquitination-dependent or deubiquitination-independent mechanisms, utilizing its papain-like protease to hinder the immune response (*Ran et al., 2022*). It is worth mentioning that children exhibit higher basal expression of RIG-1 and MDA5 in their upper airway epithelial cells, which leads to a more robust and rapid antiviral response to SARS-CoV-2 compared to adults (*Loske et al., 2022*). Children with SARS-CoV-2 infection tend to experience less severe disease compared to adults, due to a stronger innate immune response in the upper airway (*Mick et al., 2022*).

### NLRs and SARS-CoV-2

Four primary NLR types exist: AIM2, NLRP1, NLRC4, and NLRP3. Elevated NLRC4 in zebrafish has been shown to modulate the transcription of type I IFNs and interferon-stimulated genes (ISGs) by promoting the antiviral response and regulating the MDA5-MAVS and TRAF3-MAVS complexes (*Wu et al., 2020*). However, in the event that blood monocytes get infected, cytokine production and pyroptosis should result from the activation of NLRP3 and AIM2 (*Junqueira et al., 2021*). More systemic cardiovascular disease consequences are frequently associated with highly expressed NLRP1 in case of SARS-CoV-2 than with MERS-CoV or SARS-CoV (*Jha et al., 2021*).

Among these, NLRP3 is the most well-researched inflammasome, which has four parts: an LRR, a NACHT domain that binds nucleotides, a PYD and a CARD that recruits caspases. The NLRP3 inflammasome can be activated by a number of SARS-CoV PAMPs that are produced from ORF3a, ORF8b, E protein, and viral RNA (*Zhao, Di & Xu, 2021*). NLRP3 inflammasome assembly can also be triggered in human hematopoietic stem cells (HSCs) and very small embryonic-like stem cells (VSELs) by ACE2's interaction with S and N proteins of SARS-CoV-2. During SARS-CoV-2 infection, vimentin, DEAD-box helicase 3X (DDX3X), and macrophage migration inhibitory factor (MIF) are important factors that activate NLRP3 formation (*Harris & Borg, 2022*). Orally administering the NLRP3 inhibitor to SARS-CoV-2 infected hACE-K18 mice model resulted in much lower microglial inflammasome activation and a greater survival rate when compared to the untreated groups (*Lei et al., 2022*). Consequently, it is thought to be possible to use drugs to block the NLRP3 pathway and reduce cytokine release in patients (*Lei et al., 2022*).

## APOPTOTIC CELL-DEATH AND SIGNALING DURING SARS-COV-2 INFECTION

Apoptosis-dependent host cell death is an essential intrinsic antimicrobial process that limits pathogen multiplication and propagation. Inflammatory reactions initiated by SARS-CoV-2 cause apoptosis, which kills host cells (*Yuki, Fujiogi & Koutsogiannaki, 2020*). Apoptosis is believed to be involved in the pathophysiology of COVID-19 because of severe cell injury and tissue destruction of lung, kidney, liver, pancreas, neurological as well as

immunological system. It triggers the process aimed at further halting the infection, but excessive activation may lead to lung damage. The process of apoptosis can be initiated with both intrinsic and extrinsic apoptotic mechanisms. The death-inducing signaling complex (DISC), which contains Fas, Fas-associating protein with a novel death domain (FADD), and procaspase-8, is formed when extracellular ligands, such as Fas ligand (FasL), trigger cell surface death receptors (like Fas) and then activate them and activate caspase-8 (*Kaufmann, Strasser & Jost, 2012*). After that, it starts to activate caspase-3, which ultimately starts with an extrinsic apoptotic pathway. The intrinsic process (mitochondrial pathway) triggered by outside factors such DNA damage or cellular stress. Bak/Bax (BCL2 antagonist killer/BCL2 associated X protein) is activated in response to external stimuli, leading to the release of cytochrome C from mitochondria and the activation of caspase 9. Cell death of SARS-CoV-2 infected lung epithelial cells occur because of caspase 3 activation, which is initiated by activated caspase 9 (*Yuan et al., 2023*). Figure 2 illustrates the molecular mechanisms by which SARS-CoV-2 induces apoptosis in lung epithelial cells, leading to pulmonary injury. This process leads to the demise of infected cells, triggering an inflammatory response. A study reported a combination of *in vitro*, *in vivo*, and *ex vivo* models to validate the induction of apoptosis by SARS-CoV-2 and utilized MERS-CoV as a model to investigate the underlying mechanism responsible for virus-induced apoptosis (*Chu et al., 2021*). They discovered that the MERS-CoV infection's proapoptotic mediators were tightly controlled by PERK (protein kinase R-like endoplasmic reticulum kinase) signalling. The displacement of PERK from the ER chaperon GRP78 (78 KDa glucose-regulated protein) led to its activation.

ER chaperon GRP78 plays a pivotal role in numerous cellular processes, such as guiding the translocation of newly synthesized polypeptides across the ER membrane, facilitating protein folding and assembly, directing misfolded proteins for ER-associated degradation (ERAD), regulating calcium homeostasis, and acting as a sensor for ER stress. The multifaceted functions of GRP78 underscore its significance in orchestrating diverse cellular activities within the ER (*Wang et al., 2009*). Additionally, it was found that the PERK signalling converged with the intrinsic or mitochondrial apoptotic pathway. The pathogenesis of both MERS-CoV and SARS-CoV-2 was caused by the inhibition of PERK signalling and the intrinsic apoptotic pathway. Lung damage by the SARS-CoV-2 infection was lessened by PERK signalling modification (*Zhou et al., 2020*). Studies have shown that SARS-CoV-2 proteins play a significant role in apoptosis induction in several ways (*Yuan et al., 2023*). It has been established that SARS-CoV-2 ORF3a functions as a viroporin, capable of forming an ion channel on the cell membrane. This disrupts intracellular homeostasis, which plays a role in apoptosis, and facilitates virus release. By cleaving and activating caspase-8 for the extrinsic route and cross-talking to the intrinsic pathway *via* truncated BH3-interacting domain (tBID), which results in the release of cytochrome c and activation of caspase-9, a different study shown that SARS-CoV-2 ORF3a may efficiently trigger apoptosis (*Yuan et al., 2023*). In addition, it has been shown that SARS-CoV-2 ORF7b causes TNF-dependent apoptosis in HEK293T cells and Vero E6 cells (*Yang et al., 2021*). Further investigation revealed that the intrinsic and extrinsic routes caused apoptosis in a variety of cell types, including T cells, vascular endothelial cells (ECs), macrophages,

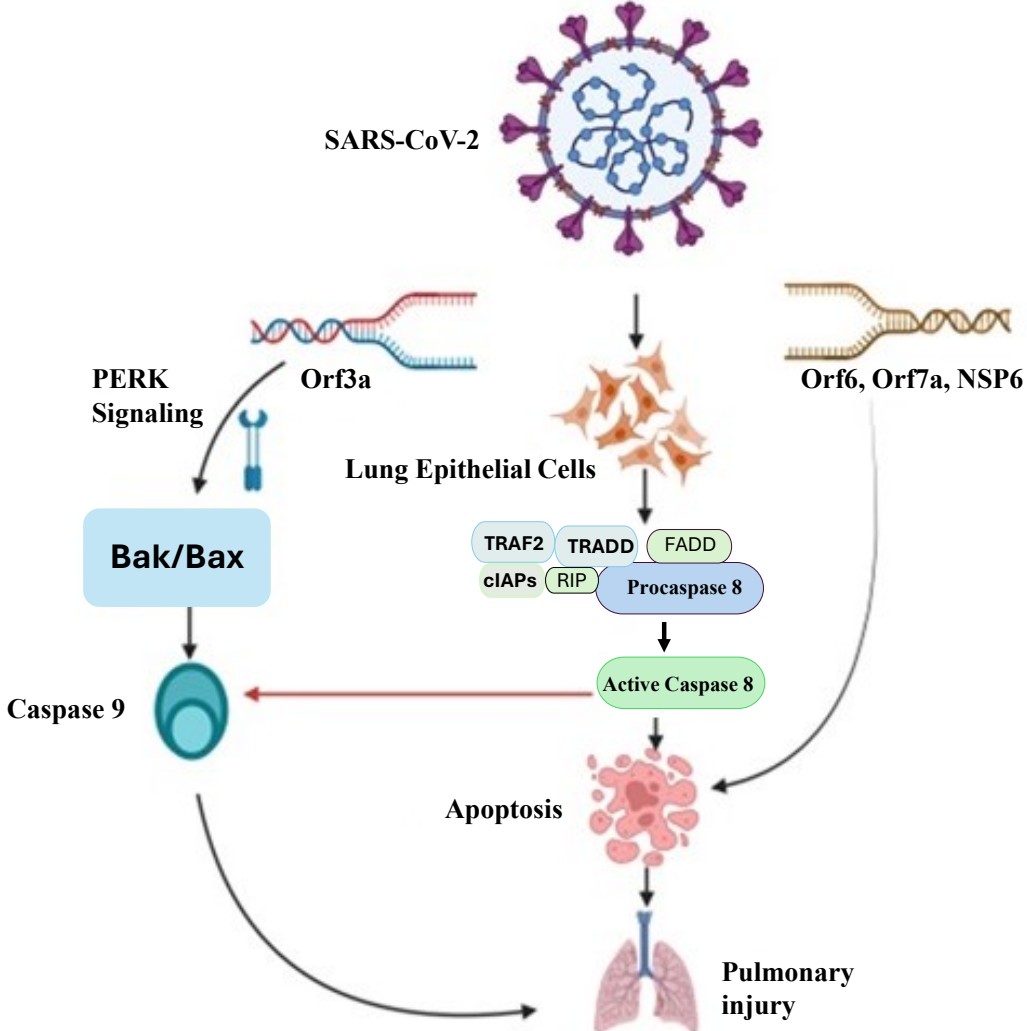

**Figure 2  Molecular mechanisms of SARS-CoV-2-induced pulmonary injury *via* apoptotic pathways.**
SARS-CoV-2 infection triggers multiple apoptotic pathways. The viral Orf3a protein activates PERK signaling, which upregulatesBak/Bax, leading to the activation ofcaspase-9 and intrinsic apoptotic pathways. Additionally, viral proteins Orf6, Orf7a, and NSP6 interact with host cellular components to activatedeath receptor-mediated extrinsic apoptosis, involving TRAF2, TRADD, FADD, RIP, and cIAPs, which facilitate the activation ofcaspase-8. Caspase-8 further promotes apoptosis by cleaving downstream effector caspases, ultimately resulting in widespreadlung epithelial cell deathandpulmonary injury. This mechanistic insight underscores the pathological impact of SARS-CoV-2 on lung tissues and highlights potential therapeutic targets to mitigate severe COVID-19 outcomes.

and alveolar type 1 and 2 cells (AT1s and AT2s) in the infected non-human primate lung. The results above are consistent with other research on the SARS-CoV-2-induced apoptosis in respiratory epithelial cells and ECs. In the animal model of SARS-CoV-2 infection, a significant portion of the renal tubular epithelial cells also undergo apoptosis, which results in acute kidney injury (AKI). Another immunological characteristic related to the severity of the SARS-CoV-2 infection is the decreased proportion of dendritic

cells (DCs) in COVID-19 patients. DCs and monocyte-derived macrophages (MDMs) were found to harm mitochondria and caspase-3 activation-dependent apoptosis, which could be stopped by anti-IFN therapy (*Li et al., 2022*). Therefore, a possible treatment for SARS-CoV-2 infection involves blocking TNF- and its receptor. These data not only show how important apoptosis is in the pathogenesis of SARS-CoV-2 but also suggest a treatment strategy that focuses on apoptosis.

## ACTIVATION AND THE SECRETION OF CYTOKINES IN SARS-COV-2 INFECTION

Cytokine markers consist of a group of polypeptides signaling molecules capable of initiating and controlling numerous cellular biological processes through the activation of cell surface receptors. Recent research has demonstrated that SARS-CoV-2 is linked with the activation of innate immunity. Based on their biological effects, the cytokine superfamily can be categorized into several main families, including interleukins (ILs), interferons (IFNs), tumor necrosis factors (TNFs), chemokines, and transforming growth factor (TGF-β). Cytokines can further be classified into pro-inflammatory or anti-inflammatory subclasses. Interleukins such as IL-1, IL-4, IL-6, IL-7, IL-10, IL-12, IL-17, and IL-18 have been demonstrated to play a significant role in the body's inflammatory response during SARS-CoV-2 infection (*Hasanvand, 2022*) and its main source are active macrophages and monocytes (*Turner et al., 2014*). Some important ILs in SARS-CoV-2 infections are described here.

### Interleukin-1

Interleukin-1 has been shown to play a major role in inflammatory response in the body against infection (*Siu et al., 2019*). Epithelial injury caused by SARSCoV-2 infection results in the secretion of IL-1α, which attracts neutrophils and monocytes towards the site of infection, and IL-1β generation in monocytes/macrophages (*Van de Veerdonk & Netea, 2020*). IL-1β forms part of the cytokine storm produced by coronavirus infections in body because of uncontrolled immune responses (*Copaescu et al., 2020*). In contrary to patients with mild COVID-19 infection, patients with severe/critical COVID-19 disease exhibited significantly greater levels of inflammatory cytokines in their bronchial alveolar lavage fluid (BALFs), particularly IL-8, IL-6, and IL-1β (*Liao et al., 2020*). One of the recognized IL-1-blocking drugs is anakinra, which functions similarly to IL-1Ra by preventing IL-1 as a result, it can stop the auto-inflammatory pathways. (*Behzadi et al., 2022*; *Van de Veerdonk & Netea, 2020*). Anakinra belongs to a class of drugs known as interleukin-1 (IL-1) receptor antagonists to treat and prevent cytokine storms thus used to relief in certain inflammatory conditions. Anakinra could offer greater benefits during the initial phases of the ailment, before elevated cytokine levels emerge, potentially thwarting the advancement to critical illness and the need for mechanical ventilation (*Khani et al., 2022*). However, in other studies it could not provide notable improvement (*Dahms et al., 2023*; *Elmekaty et al., 2023*). Further research on IL-1 inhibitor agents to mitigate the inflammatory consequences initiated by SARS-CoV-2 infection may be a promising approach to control innate immune response (*Mardi et al., 2021*).

### Interleukin-4

Multiple investigations involving SARS-CoV-2 patients have identified increased IL-4 levels, contributing to the cytokine storm linked to severe respiratory symptoms (*Hu, Huang & Yin, 2021*; *Liu et al., 2020*). The activation and release of IL-4, leading to the stimulation of the IL-4 receptor, which inhibits the secretion of various inflammatory cytokines such as TNF-α, IL-1, and PGE2. IL-4 is secreted from various immune cells, including T helper type 2 (Th2) cells. These interleukins cause Th2 cells to respond by blocking the Th1 immune response pathway. It was discovered that there is an increased production of Th1 cells in situations of overactive immunity responses and autoimmune diseases. Th2 cells were observed to be more prevalent in COVID-19 patients receiving high-intensity treatment (*Montazersaheb et al., 2022*). Findings and data gathered during the COVID-19 pandemic reveal a notable increase in Th2, Th1/Th17 cells, and antibody production in COVID-19 patients. Additionally, it has been demonstrated that Th2 cells can activate interleukin 4, which then triggers apoptosis by activating the STAT signalling pathway (*Renu et al., 2020*).

### Interleukin-6

In SARS-CoV-2 infection, there is noted elevation in the secretion or production of IL-6 and IL-8 in patients, coupled with a reduction in CD4+ and CD8+T cells (*Rabaan et al., 2021*; *Zheng et al., 2020*). A research conducted by Ruan et al. indicates that levels of IL-6 and ferritin were elevated in patients who succumbed to SARS-CoV-2 compared to those who recovered (*Ruan et al., 2020*). Research indicates that individuals with hypertension, elevated levels of IL-6, and SARS-CoV-2 infection are at a significantly increased risk of developing severe respiratory failure (*Zhang et al., 2020a*). One of the most promising approaches in managing cytokine storms in SARS-CoV-2 patients could involve the inhibition of the IL-6 receptor using tocilizumab, aiming to avert severe complications from the virus (*Pelaia et al., 2021*). Utilizing IL-6 receptor blockers stands out as one of the highly recommended treatments for SARS-CoV-2, offering a promising avenue for intervention (*Hasanvand, 2022*). Abnormalities in innate lymphoid cells have been detected in patients with SARS-CoV-2 infection (*Kumar et al., 2021a*), this could be linked to disorders in the IL-7 signalling pathway and its receptor (*Sheikh & Abraham, 2019*). There is a suggestion that IL-7 might have potential applications as a vaccine adjuvant hence augmenting immune responses to vaccines particularly those targeting SARS-CoV-2 or other emerging pathogens (*Bekele, Sui & Berzofsky, 2021*).

### Interleukin-10

Studies has indicated a substantial increase in serum interleukin-10 (IL-10) levels during the cytokine storm observed in patients with COVID-19 infection (*Huang et al., 2020*). Increased serum interleukin-10 levels in COVID-19 patients may serve as both an anti-inflammatory mechanism and an immunosuppressive biomarker (*Islam et al., 2021*). Studies have shown that recombinant IL-10 can be utilized to exhibit anti-fibrotic activity and modulate immune-regulating functions in patients with COVID-19 (*Lu et al., 2021b*).

### Interleukin-17

Recent reports suggest that IL-17 plays a role in the hyperinflammatory state seen in COVID-19 . The increased expression of IL-17A during the cytokine storm is attributed to T helper 17 cells and is primarily linked to ARDS. Consequently, there has been a proposal for the potential therapeutic application of IL-17 inhibitors in COVID-19 (*Maione et al., 2021*).

### Interleukin-18

IL-18 is a proinflammatory cytokine that appears to play a role in the cytokine storm and hyperinflammation associated with severe COVID-19 cases. Interleukin (IL)-18 serves as a pivotal cytokine in macrophage activation syndrome. Elevated levels of IL-18 have been observed in COVID-19 patients, suggesting it could be a potential therapeutic target (*Satış et al., 2021*).

### Tumour necrosis factor

Studies have shown that during SARS-CoV-2 infection, tumour necrosis factor (TNF)R1 expression is increased (*McElvaney et al., 2020*) and increased serum TNF-α levels in these patients are associated with increased disease severity (*Chen et al., 2020a*; *Leija-Martínez et al., 2020*). Studies have demonstrated that during SARS-CoV-2 infection, the expression of sTNFR1 is elevated in COVID-19 patients (*McElvaney et al., 2020*) Conversely, research indicates that serum TNF-α levels are increased in these patients and are correlated with high disease severity (*Qin et al., 2020*) .

### Transforming growth factor

Complications resulting from transforming growth factor (TGF-β) secretion in individuals with SARS-CoV-2 infection may include the initiation of interstitial lung alterations, increased pulmonary secretions, sputum production, dry cough, bronchial asthma, and ultimately, disruption of regular breathing patterns (*Costela-Ruiz et al., 2020*). In cases of SARS-CoV-2 infection, examination of TGF-β levels indicates an increase in serum levels of this cytokine among patients, consequently resulting in the suppression of immune system activity in those who are infected (*Ferreira-Gomes et al., 2021*).

### Secretion of interferon

Interferons (IFNs) are signalling proteins that play a crucial role in the immune response to viral infections and other immune challenges. They have several important functions in the immune system: antiviral defence, immune modulation, inflammation regulation, enhancing antigen presentation, immune surveillance, Immunostimulatory and immunomodulatory therapies (*Danladi & Sabir, 2021*; *Mihaescu et al., 2023*). IFN-γ stands as a pivotal cytokine released by both NK cells and T lymphocytes, holding a crucial position in enhancing the body immune response. In the context of cytokine storms associated with SARS-CoV-2 infection, anomalies in IFN-γ levels become evident, accompanied by an overexpression of genes associated with COVID-19 (*Gadotti et al., 2020*). Hub genes identified in the disease-gene interaction network play a critical role in regulating the immune response during COVID-19 infection. Hub genes exhibited a close

association with the activation of CD4 memory T cells, regulatory T cells, and activated NK cells and genes BIRC5, DNAJC4, DTL, LILRB2, and NDC80 were identified as having robust diagnostic properties as well (*Zhou et al., 2023*).

## Chemokines in SARS-CoV-2 infection

A family of small cytokines called chemokines plays a critical role in mediating proper immune responses. Chemokines play a significant role in recruiting phagocytes to the site of infection, and they have important functions in both diagnostics and therapeutics, like in COVID-19 patients, the chemokines CXCL1, CXCL3, CXCL6, CXCL15, CXCL16, and CXCL17 have been associated with the recruitment of macrophages (*Schultze & Aschenbrenner, 2021*). These chemokines play an important role in the immune response to SARS-CoV-2. Understanding the role of chemokines in macrophage recruitment is important in COVID-19 research and treatment.

### CCL2/MCP-1

The chemokine (C-C motif) ligand 2 (CCL2/ also known as monocyte chemoattractant protein 1 (MCP1) and its cognate receptor (CCR2) are unregulated in COVID-19 patients (*Bagheri et al., 2024*) and is linked to predict the severity of disease. CCL2 is secreted during the early phase of infection and is significantly increased further during late stages of fatal cases than severe and/or mild COVID-19 patients (*Xu et al., 2020*). CCL2 involved in the recruitment of monocytes and macrophages to sites of infection and plays a vital role in immune response to viral challenges (*Bagheri-Hosseinabadi et al., 2024*). In the lungs, CCL2 is mainly produced by alveolar macrophages, T cells and endothelial cells and CCR2 is mainly expressed on monocytes and T cells (*Henrot et al., 2019*). Increased levels of CCL2 in bronchoalveolar lavage fluid (BALF) are associated with initiating a cytokine storm and promoting the accumulation of CD163+ myeloid cells in the airways and further alveolar damage in the lungs of patients with SARS-CoV-2 (*Ranjbar et al., 2022*). Moreover, increased CCL2 levels were reported to be correlated with the development of respiratory failure (*Jøntvedt Jorgensen et al., 2020*) and acute kidney injury in critically ill COVID-19 patients (*Bülow Anderberg et al., 2021*).

### CCL3

Similar to CCL2, CCL3 plays a vital role in the recruitment and activation of monocytes and macrophages, including T cells to sites of infection and It involved in immune response and the regulation of antiviral defense (*Trifilo et al., 2003*). *Abers et al. (2021)* rreported that increased serum CCL3 level was directly associated with the mortality rate of patients with COVID-19. A study demonstrated that higher serum concentrations of CCL2/MCP-1, CCL3/MIP-1a, and CCL5/RANTES was observed in COVID-19 patients and these cytokines play an important role in causing inflammatory complication (*Hu, Huang & Yin, 2021*).

### CCL5

CCL5 is a chemotactic cytokine that activates immune cells in the peripheral immune system during acute viral infection (*Crawford et al., 2011*; *Maghazachi, Al-Aoukaty &*

*Schall, 1996*). Researchers reported that increased expression of CCL5 helps to eliminate SARS-CoV-2 infection and prevent the severity of disease (*Zhao et al., 2020*). However, other studies reported that higher levels of CCL5 are associated with liver and kidney injuries (*Chen et al., 2020b*; *Yu et al., 2016*) and a study also observed an elevated CCL5 levels in SARS-CoV-2 infected patients with liver and kidney injuries compared to healthy controls or mildly and moderately (*Patterson et al., 2020*).

## cGAS-STING SIGNALING PATHWAY

The cGAS-STING signaling pathway is a critical component of the innate immune response that helps, detect and defend against viral infections and DNA pathogens (*Ahn & Barber, 2019*).The innate immune system is the body's initial defence against invading pathogens (*Akira, Uematsu & Takeuchi, 2006*). It identifies specific patterns found in pathogens or damaged cells using PPRs, which include TLRs, Nod-like receptors (NLRs), RIG-I-like receptors (RLRs), and the DNA sensor cyclic guanosine monophosphate (GMP)-adenosine monophosphate (AMP) synthase (cGAS)-stimulator of interferon genes (STING) signaling pathway. Among these receptors, the cGAS-STING pathway, plays a significant role in the innate immune response to viral infections. Multiple lines of evidence have suggested that the cytoplasmic DNA sensor cGAS-STING not only recognizes dsDNA viruses but also plays a crucial role in RNA virus infection, either by directly recognizing virus characteristics or by detecting cellular DNA released from mitochondria or nuclei in response to cellular stress (*Li et al., 2021*) (Fig. 3). STING has been linked to SARS-CoV-2 infection by triggering the type I IFN response (*Chattopadhyay & Hu, 2020*; *Messaoud-Nacer et al., 2022*; *Ramanjulu et al., 2018*). Although type I IFNs are quickly induced to stop the spread of the virus, a sustained rise in type I IFN levels during the late stages of the infection is linked to aberrant inflammation and a poor clinical outcome (*Liu et al., 2021*). The cGAS-STING pathway is demonstrated to be a major regulator of aberrant type I IFN responses in COVID-19 (*Humphries et al., 2021*). Application of a STING inhibitor reduced STING activation, lowering the severe lung inflammation brought on by SARS-CoV-2 and improving the course of the disease (*Chattopadhyay & Hu, 2020*; *Liu et al., 2021*). In SARS-CoV-2 infected people and mice, activation of cGAS-STING results in a rise in inflammation and pathogenesis (*Liu et al., 2021*). STING inhibitors can limit this response, according to *Neufeldt et al. (2022)* discovery that SARS-CoV-2 infection activates the cGAS-STING route, which causes the production of proinflammatory cytokines mediated by the nuclear factor B (NF-B) pathway (*Li et al., 2021*). Both *Neufeldt et al. (2022)* observation of STING colocalization with SARS-CoV-2 N protein in infected cells and *Rui et al. (2021)* observation of interactions between STING and ORF3a suggest viral proteins have a direct role in modifying the cGAS-STING pathway (*Li et al., 2021*; *Liu et al., 2022*). According to data from an earlier investigation, STING activation is a potential therapeutic strategy to manage SARS-CoV-2 (*Li et al., 2021*) (Table 1). Recent research has demonstrated that STING agonists influence the type I IFN response, which in turn affects SARS-CoV-2 infection (*Chattopadhyay & Hu, 2020*; *Li et al., 2021*) and utilization of cGAS-STING pathway agonists holds a promise for a vaccine adjuvants (*Tian*

*et al., 2024*). *Li et al. (2021)* performed high-throughput screening to find antiviral innate immune agonists to prevent SARS-CoV-2 infection, and they discovered endogenous STING agonists, cyclic dinucleotides (CDNs), as antiviral drugs against SARS-CoV-2. Strong small molecule STING agonists, including diABZI, have been exploited because of limited potency of CDNs and poor drug quality (*Liu et al., 2021*; *Messaoud-Nacer et al., 2022*; *Ramanjulu et al., 2018*). *Li et al. (2021)* studied the small chemical STING agonist diABZI and observed that it can successfully prevent SARS-CoV-2 infection of several strains by activating IFN signalling. Notably, diABZI can inhibit viral replication in live mice and primary human bronchial epithelial cells. Consequently, this STING agonist may be applied as a novel treatment approach to combat COVID-19. Like this, *Humphries et al. (2021)* reported a diamidobenzimidazole drug diABZI-4, which stimulates STING and is particularly effective in limiting SARS-CoV-2 replication in cells and mice. When diABZI-4 was administered intravenously, STING was activated quickly, which helped to temporarily increase the production of proinflammatory cytokines and activate lung lymphocytes and inhibit viral replication (*Humphries et al., 2021*). There are a number of new cGAS-STING activators, including colloidal manganese salt, CF501, mucoadhesive nanoparticles, and IAPA (indirect-acting pan-antiviral) agents, which offer fresh perspectives on anti-SARS-CoV-2 therapy (*Jearanaiwitayakul et al., 2022*; *Kleandrova, Scotti & Speck-Planche, 2021*; *Liu et al., 2022*; *Zhang et al., 2021b*). The stability of STING agonists has improved with the development of the biocompatible peptide, protein, and bio membrane platforms (*Zheng & Wu, 2022*). Few adjuvants have been approved for use in humans thus far, and the only one that is often used is one that contains aluminum (*Clapp et al., 2011*). Nevertheless, it has been demonstrated that the nanoparticle manganese (nanoMn) adjuvant, also known as STING agonists, promotes antigen presentation, virus-specific memory T-cell generation, and host-adaptive immunity, making it the ideal adjuvant for protein-based COVID-19 subunit vaccines (*Wu et al., 2021*) (Table 2). The nanoMn adjuvant, according to *Zhang et al. (2022)* is the most efficient at boosting the immunogenicity or immune responses of SARS-CoV-2 protein-based subunit vaccines. Additionally, the use of a new STING agonist, CDGSF, in combination with the SARS-CoV-2 S protein as an adjuvant result in extraordinarily high antibody titres and a potent T-cell response, outperforming the drawbacks of adjuvants that contain aluminum (*Wu et al., 2021*). NanoSTING, used as an adjuvant for intranasal vaccination with S protein trimeric or monomeric form, evoked potent serum neutralizing antibodies and T-cell responses (*Berthelot & Lioté, 2020*). Strong stimulatory effects on antibody responses in the respiratory tract were seen when the S protein and cGAMP were administered (*Berthelot & Lioté, 2020*; *Chauveau et al., 2021*).

According to the data, STING agonists significantly increased the S protein immunogenicity (*Zhang et al., 2021b*). These results highlighted the STING agonist adjuvant potential in the SARS-CoV-2 vaccination. STING agonists, which instantly enhance IFNs signalling, can quickly and transiently activate STING (*Humphries et al., 2021*; *Li et al., 2021*). Nevertheless, it is important to prevent the progression of the illness brought on by excessive inflammation when using STING vaccination adjuvants that induce long-lasting humoral and cellular immune responses (*Liu et al., 2022*). As a result, inhibiting STING activation reduces inflammatory responses and pathogenesis,

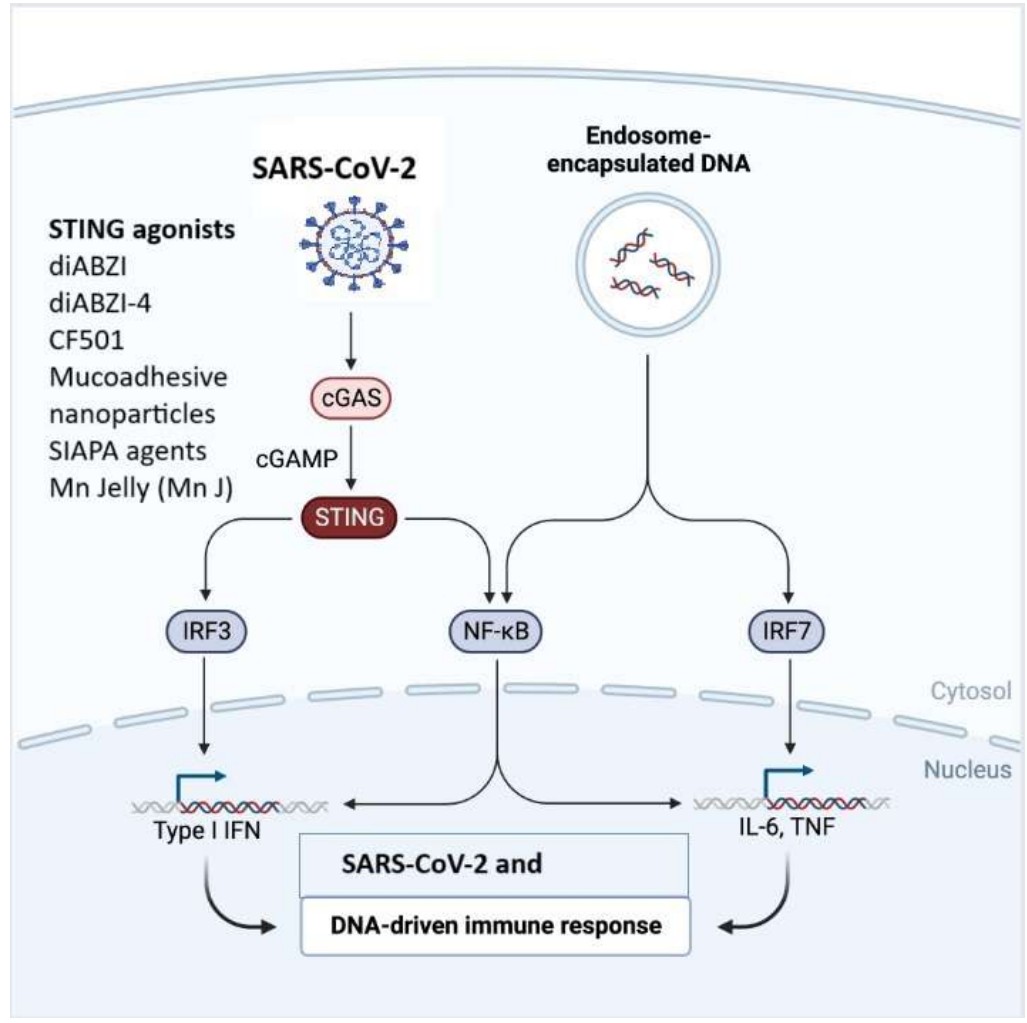

**Figure 3** **The cGAS-STING signaling pathway in immune responses to SARS-CoV-2 and DNA.** SARS-CoV-2 or endosome-encapsulated DNA, activates cyclic GMP-AMP synthase (cGAS), leading to the production of cyclic GMP-AMP (cGAMP). cGAMP binds to the stimulator of interferon genes (STING), triggering a signaling cascade. STING activation recruits and stimulates transcription factors, including IRF3, NF-κB, and IRF7, resulting in the production of type I interferons (IFNs) and pro-inflammatory cytokines such as IL-6 and TNF. This pathway is essential in mounting an innate immune response to viral infections and cytosolic DNA. The figure illustrates STING agonists, including diABZI diacylbenzimidazole), diABZI-4, CF501, mucoadhesive nanoparticles, SIAPA (stimulator of interferon genes-activating polymeric agents), and Mn Jelly (manganese jelly), which enhance antiviral immunity by activating the cGAS-STING pathway.

indicating that STING may be exploited as a therapeutic target to prevent SARS-CoV-2-related severe illness symptoms. Incorporating STING activation into antibody therapy for managing SARS-CoV-2 involves designing STING agonist antibodies to enhance type I interferon response within infected cells, strengthening immunity against the virus. Neutralizing antibodies could block viral components that hinder the STING pathway, indirectly promoting STING activation. Combination therapies combining

**Table 1  Potential therapeutic strategies for targeting STING in SARS-CoV-2 infection.**

| S. No. | Name | Therapeutic strategy | Description | References |
|--------|------|---------------------|-------------|------------|
| 1. | diABZI | STING agonist | Suppresses SARS-CoV-2 infection | *Li et al. (2021)* |
| 2. | diABZI-4 | STING agonist | Stimulates STING and is highly effective in inhibiting SARS-CoV-2 replication | *Humphries et al. (2021)* |
| 3. | CF501 | STING agonist | Vaccine to resist the SARS-CoV-2 and its variants | *Liu et al. (2022)* |
| 4. | Mucoadhesive nanoparticles | STING agonist | The intranasal delivery system loaded with cGAMP potently boost the immunogenicity of the spike based vaccine in the respiratory tract | *Jearanaiwitayakul et al. (2022)* |
| 5. | IAPA agents | STING agonist | Boosts the immune system through STING, while blocking essential inflammation-related proteins such as caspase-1 and TNF-α | *Kleandrova, Scotti & Speck-Planche (2021)* |
| 6. | Mn Jelly (Mn J) | STING agonist | Mn J made to serve not only as an immune enhancer but also as a delivery system to activate humoral and cellular immune response | *Zhang et al. (2021d)* |

**Table 2  Potential vaccine adjuvants targeting stimulator of interferon genes (STING) in SARS-CoV-2 infection.**

| S. No. | Name | Therapeutic strategy | Description | References |
|--------|------|---------------------|-------------|------------|
| 1 | NanoSTING | Vaccine adjuvant | NanoSTING as the adjuvant for intranasal vaccination of S protein trimeric or monomeric form | *Li et al. (2021)* |
| 2 | cGMP-ternary adjuvant | Vaccine adjuvant | A novel ternary adjuvant system with alum/STING agonist 3,3′-cGAMP/poly(I:C) | *Liu et al. (2022)* |
| 3 | NanoMn | Vaccine adjuvant | Enhances cellular uptake and sustained release of $Mn^{2+}$ in a pH-sensitive manner, thereby enhancing IFN response | *Sun et al. (2021)* |
| 4 | CDG[SF] | Vaccine adjuvant | CDG[SF] as an adjuvant immunisation with SARS-CoV-2 S protein | *Wu et al. (2021)* |

STING-targeting antibodies with antiviral drugs or mAbs could provide a comprehensive. Engineered antibodies might also bolster broader immune responses, such as T cell activation, while others could finely modulate STING-triggered inflammation to prevent excessive reactions. Conjugating antibodies with STING agonists or immune modulators allow targeted delivery to infected cells, enhancing efficacy and minimizing side effects. Personalized antibody therapies, adjusted to individual immune profiles, offer potential for optimized treatment outcomes.

## SARS-COV-2 PROTEINS AND THEIR INNATE IMMUNE TARGETS

SARS-CoV-2 proteins show varying degrees of target tropism and have diverse functions. The role of each SARS-CoV-2 protein in innate immune response was mentioned in details in Table 3 (*Minkoff & tenOever, 2023*). SARS-CoV-2 proteins nsp-6 and nsp-13 bind TANK binding kinase-1 (TBK-1), leading to inhibition of IRF-3 phosphorylation and subsequent reduction in IFN-beta production. Nuclear translocation of IRF-3 is also abrogated by binding of ORF-6 to importin karyopherin alpha-2 (*Lei et al., 2020*; *Schreiber, 2020*) eventually leading to downregulation of type-1 IFN secretion. The abrogation of IRF-3 nuclear translocation and downregulation of type-1 IFN expression can also be from the antagonistic nature of viral proteins ORF3b or due to M protein-dependent ubiquitin-mediated degradation of TBK1 (*Konno et al., 2020*; *Sui et al., 2021*). NF-kB and IFN-beta pathway activation is inhibited by ORF6, ORF8, and N proteins of SARS-CoV-2. Moreover, ORF6, ORF8, and nsp-1 abandon the ISRE-driven transcription of ISGs (*Li et al., 2020*). Researchers identified interactions between ISGs and TLR3 agonists, including poly I:C and imiquimod. This discovery suggests a potential for drug repurposing in the realm of COVID-19, proposing TLR3 agonists as promising candidates for therapeutic exploration. ISGs, such as IFIT and IFITM, ISG15, IFIH1, MX1, IRF7, OAS 1-3, and STAT1 are recognized for enhancing IFN signaling, thereby contributing to antiviral activity, emerge as potential candidates for drug targets in COVID-19 treatment (*Prasad et al., 2020*). Some other proteins involved in the downregulation of type-1 IFN expression are ORF-96, nsp13, nsp-1, and M proteins, and the target of these viral proteins are RIG-1/MDA-5/MAVS signaling cascade (*Jiang et al., 2020*; *Ricci et al., 2021*).

The phosphorylation of STAT-1 and STAT-3 is inhibited by SARS-CoV-2 viral proteins including nsp-6, nsp-13, ORF-39, and ORF-7b, so these proteins have an antagonistic impact on the type-1 IFN signaling pathway (*Li et al., 2020*). In addition, the prevention of STAT phosphorylation and their nuclear translocation is suppressed by other SARS-CoV2 proteins like N, ORF-6, and M proteins (Fig. 4) (*Ricci et al., 2021*; *Schultze & Aschenbrenner, 2021*). A study shows that patients with inborn errors in TLR-3 and IRF-3 dependent type-1 IFN immune response presented with severe SARS-CoV-2 infection, concluding the vital role of type-1 IFN in combating the viral infection (*Zhang et al., 2020b*). In another study, 3.5% of patients with autosomal recessive deficiencies in IRF-7 and IFNAR1 genes and autosomal dominant deficiencies in genes encoding TLR-3, unc-93 homolog B1, TLR adaptor molecule 1, TBK1, IRF-3, IRF-7, IFNAR1, and IFNAR2 showed severe COVID-19 pneumonia (*Gao et al., 2020*). The role of type-1 IFN is demonstrated by a study where the anti- type1 IFN autoantibodies have been observed in severe COVID-19 infections (*Bastard et al., 2020*). The nonstructural protein 16 (nsp16) derived from SARS-CoV-2 diminishes the splicing of overall mRNA and hinders the identification of viral RNA by intracellular helicase receptors. Additionally, nsp-1 disrupts mRNA translation by attaching to 18s ribosomal RNA within the mRNA entry channel, while both NSP-8 and NSP-9 impede protein trafficking to the cell membrane. These three distinct mechanisms collectively exert

**Table 3 SARS-CoV-2 proteins as innate immune targets.**

| SARS-CoV-2 proteins, (amino acid length) | Role in innate immunity | References |
|---|---|---|
| **Structural proteins** | | |
| Spike (S), (1273 aa) | Enhances the proteasomal degradation of IRF3 and disrupts host sensor recognition, block IFN signalling by blocking the interaction between STAT1 and JAK1, activates NF-κB *via* promoting the phosphorylation of p65 and IκBα | *Freitas, Crum & Parvatiyar (2021)*, *Olajide et al. (2022)*, *Zhang et al. (2021d)* |
| Membrane (M) (222 aa) | Blocks host sensor recognition, Inhibits MAVS activation by disrupting its capacity to form essential large aggregates for the recruitment of signalling adaptors, decreases TBK1 expression through ubiquitin-mediated degradation, obstructs nuclear transport by binding to KPNA6 importin, thereby preventing its interaction with IRF3. | *Fu et al. (2021)*, *Sui et al. (2021)*, *Zhang et al. (2021d)* |
| Nucleocapsid (N) (419 aa) | Mask inflammatory RNA, achieves this by binding to and destabilizing dsRNA, Possesses inherent RNA-binding properties due to its involvement in virion assembly, blocks host sensor recognition achieved by preventing the formation of stress granules through binding and sequestering the G3BP1 nucleating protein, hinders the interaction between RIG-I and TRIM25 by binding to the DExD/H box RNA helicase domain of RIG-I, inhibits the polyubiquitination and aggregation of MAVS, potentially through liquid-liquid phase separation (LLPS). | *Caruso et al. (2021)*, *Chen et al. (2020c)*, *Cubuk et al. (2021)*, *Gori Savellini et al. (2021)*, *Lu et al. (2021a)*, *Wang et al. (2021a)*, *Zheng et al. (2022)* |
| **Non-structural proteins** | | |
| Nsp1 (180 aa) | Inhibits host sensor recognition by preventing IRF3 phosphorylation, Suppresses IFN signaling pathways by reduces TYK2 and STAT2 levels, impairs nuclear transport by interacting with the mRNA export receptor heterodimer NXF1–NXT1, halts translation processes, facilitates the degradation of cellular mRNA that lacks the 5′ viral leader sequence, obstructs mRNA entry into the ribosome by binding to specific domains within the C terminus | *Banerjee et al. (2020)*, *Finkel et al. (2021)*; *Kumar et al. (2021b)*, *Lapointe et al. (2021)*, *Schubert et al. (2020)*, *Thoms et al. (2020)*, *Yuan et al. (2020)*, *Zhang et al. (2021e)* |
| Nsp3 (PLpro) (1945 aa) | Masks inflammatory RNA that essential for the creation of ER-associated DMVs, prevents detection by host sensors, cleaves IRF3 thereby disrupting its function, compromises the functionality of host proteins, macrodomain-X interacts with amino acid chains, hydrolyzing the ADP-ribose bond, PLpro domain deubiquitinates and deISGylates host signaling protein substrates. | *Alhammad et al. (2021)*, *Frick et al. (2020)*, *Klemm et al. (2020)*, *Liu et al. (2021a)*, *Michalska et al. (2020)*, *Moustaqil et al. (2021)*, *Ricciardi et al. (2022)*, *Shin et al. (2020)* |

**Table 3** (*continued*)

| SARS-CoV-2 proteins, (amino acid length) | Role in innate immunity | References |
|---|---|---|
| Nsp4 (500 aa) | Diminishes the presence of inflammatory RNA that essential for the creation of endoplasmic reticulum-associated double-membrane vesicles (ER-associated DMVs) | *Ricciardi et al. (2022)* |
| Nsp5 (3CLpro) (306 aa) | Blocks host sensor recognition, hampers the assembly of stress granules, cleave the N-terminal domain of RIG-I and hindering its interaction with MAVS, encourages the ubiquitination and subsequent degradation of MAVS, cleaves IRF3 thus impeding its function, blocks the nuclear translocation of IRF3, prevents the phosphorylation and activation of NF-κB by cleaving TAB1 and NEMO | *Chen et al. (2022)*, *Fung et al. (2021)*, *Liu et al. (2021b)*, *Moustaqil et al. (2021)*, *Zhang et al. (2021c)*, *Zheng et al. (2022)* |
| Nsp6 (290 aa) | Minimizes or blocks inflammatory RNA by attaches DMVs to the ER, hinders host sensor recognition by binding to TBK1, preventing its activation through phosphorylation, Impedes IFN signaling pathways, suppresses the phosphorylation of STAT1 and STAT2 | *Ricciardi et al. (2022)*, *Xia et al. (2020)* |
| Nsp8 (198 aa) | Halts translation process *via* attaches to the 7SL RNA scaffold element within the SRP complex | *Banerjee et al. (2020)* |
| Nsp9 (198 aa) | Disrupts nuclear transport *via* reduced expression of Nup62 on the nuclear envelope, halts translation, associates with the 7SL RNA scaffold element of the SRP complex, obstructing its capacity to bind SRP19, which is crucial for the correct folding and assembly of SRP | *Banerjee et al. (2020)*, *Gordon et al. (2020)* |
| Nsp10 (139 aa) | Reduces or conceals inflammatory RNA, functions as a co-factor alongside Nsp14 and Nsp16 in viral capping, disrupts the translation process, amplifies Nsp14-mediated translational inhibition | *Hsu et al. (2021)*, *Krafcikova et al. (2020)*, *Silva et al. (2021)*, *Wilamowski et al. (2021)*, *Yang et al. (2021)* |
| Nsp12 (RdRp) (932 aa) | Minimizes inflammatory RNA, serves as a guanylyl transferase in the process of viral mRNA capping, blocks host sensor recognition, hinders the nuclear translocation of IRF3 | *Walker et al. (2021)*, *Wang et al. (2021b)*, *Yan et al. (2021)* |
| Nsp13 (helicase) (596 aa) | Diminishes inflammatory RNA, exhibits 5′ RNA triphosphatase activity during viral mRNA capping, disrupts host sensor recognition *via* binding to TBK1 and inhibiting its activation through phosphorylation blocks IFN signalling, leading to a decrease in endogenous IFNAR1 levels, prevent phosphorylation of STAT1 and STAT2. | *Fung et al. (2022)*, *Hayn et al. (2021)*, *Vazquez et al. (2021)*, *Walker et al. (2021)*, *Xia et al. (2020)* |

**Table 3** (*continued*)

| SARS-CoV-2 proteins, (amino acid length) | Role in innate immunity | References |
|---|---|---|
| Nsp14 (Exon) (527 aa) | Minimizes inflammatory RNA, performs N7-methyltransferase activity as part of viral mRNA capping, block IFN signalling, directs IFNAR1 towards lysosomal degradation, activates NF-κB, leading to enhanced nuclear translocation of p65 and the upregulation of pro-inflammatory chemokines such as IL-6 and IL-8 | *Hayn et al. (2021)*, *Hsu et al. (2021)* |
| Nsp15 (346 aa) | Mask inflammatory RNA, utilizes endoribonuclease activity to cleave 5′-polyuridines from the negative strand of viral RNAs thus decreasing the accumulation of viral PAMPs, blocks nuclear transport, engages with the host's nuclear transport machinery specifically interacting with nuclear transport factor 2. | *Frazier et al. (2021)*, *Gordon et al. (2020)* |
| Nsp16 (298 aa) | Minimizes inflammatory RNA, demonstrates 2′-O-methyltransferase activity as part of viral mRNA capping, halts translation *via* attaches to the mRNA recognition domains found in snRNA U1 and U2 subunits of the spliceosome. | *Wilamowski et al. (2021)*, *Banerjee et al. (2020)* |
| ORF3a (275 aa) | Blocks IFN signalling *via* preventing phosphorylation of STAT1 | *Xia et al. (2020)* |
| ORF3b/3d (154 aa) | Blocks recognition by host sensors *via* preventing nuclear translocation of IRF3 | *Olajide et al. (2022)* |
| ORF6 (61 aa) | Blocks nuclear transport by binding to karyopherin-α2 (KPNA2) importin, attaches to the Nup98–Rae1 complex and preventing their interaction with the nuclear pore complex (NPC), facilitates the nuclear accumulation of host mRNAs and mRNA transporters | *Addetia et al. (2021)*, *Gordon et al. (2020)*, *Gori Savellini et al. (2022)*, *Kato et al. (2021)*, *Kawai & Akira (2009)*, *Kimura et al. (2021)*, *Miorin et al. (2020)*, *Xia et al. (2020)* |
| ORF7a 121 aa) | Blocks host sensor recognition by lowers TBK1 expression, blocks IFN signalling by preventing the phosphorylation of STAT1 and STAT2. | *Kouwaki et al. (2021)*, *Xia et al. (2020)* |
| ORF7b (44 aa) | Prevents host sensor recognition, disrupts RIG-I and MDA5 signalling through a MAVS-dependent mechanism, blocks IFN signalling by inhibiting the phosphorylation of STAT1 and STAT2. | *Kouwaki et al. (2021)*, *Shemesh et al. (2021)*, *Xia et al. (2020)* |
| ORF8 (121 aa) | Induces NF-κB activation, acts as a viral mimic of IL-17A, prompting the heterodimerization of the human IL-17 receptor and subsequent activation of NF-κB downstream pathways. | *Wu et al. (2022)* |

| SARS-CoV-2 proteins, (amino acid length) | Role in innate immunity | References |
|---|---|---|
| ORF9b (97 aa) | Blocks host sensor recognition, hinders the interaction between RIG-I and MAVS, attaches to TOM70, disrupting the TOM70/HSP90 interaction, interfering with TBK1/IRF3 signalling, blocks TBK1 phosphorylation by preventing the interaction between TBK1 and TRIF. | *Brandherm et al. (2021), Gao et al. (2021), Han et al. (2021), Jiang et al. (2020); Kouwaki et al. (2021)* |

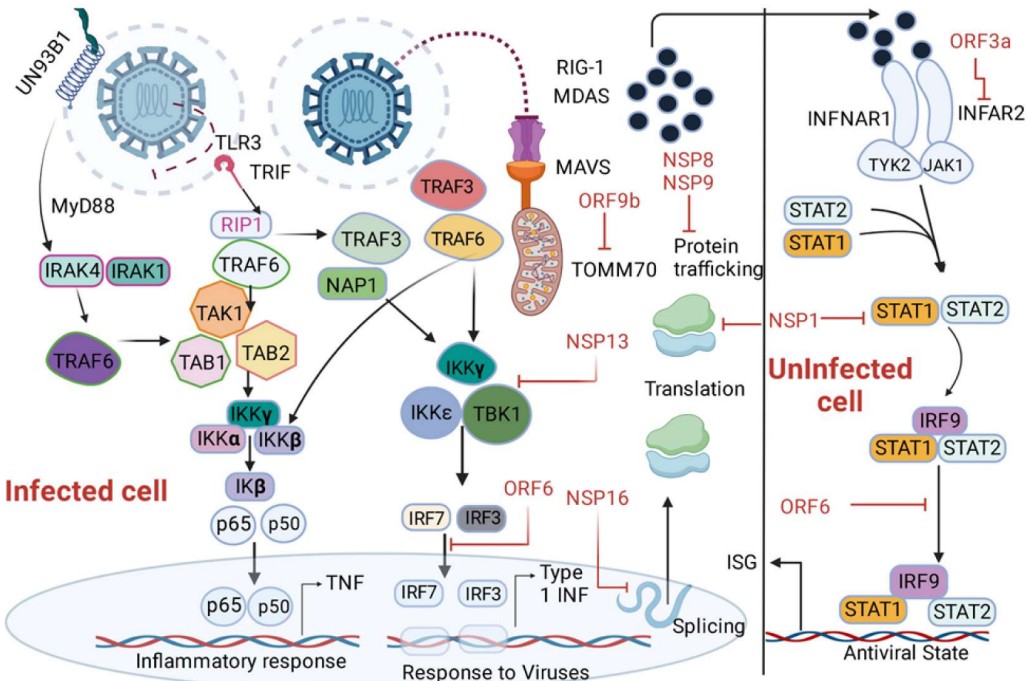

**Figure 4** **Depiction of the innate immune signalling pathway, focusing on RIG-I/MDA-5/MAVS and their inhibition by SARS-CoV-2 proteins.** The pathway begins with the recognition of viral double-stranded RNA (dsRNA) by RIG-I-like receptors, including RIG-I and MDA-5, triggering a cascade of immune responses. Activation of MAVS (mitochondrial antiviral signalling protein) leads to downstream signalling involving MyD88, IRAK1/4, and NAP1, resulting in the activation of transcription factors such as IRF3, IRF7, IRF9, and NF-κB subunits (p50/65). This activation promotes the production of interferons (*via* IFNAR1 signalling) and interferon-stimulated genes (ISGs), critical for antiviral responses. The figure also highlights SARS-CoV-2 protein inhibitors, including non-structural proteins (NSPs) and open reading frames (ORFs), which interfere with various signalling components, such as MAVS and IRF pathways, to suppress immune activation. Additionally, key molecules like angiotensin-converting enzyme 2 (ACE2), neuropilin 1 (NRP1), and receptor-interacting serine/threonine kinase 1 (IkB) kinases (IKKα/β/γ/ε) are illustrated, emphasizing their roles in the pathway.

detrimental effects on the production of type-1 interferon by the infected cells (*Banerjee et al., 2020*).

Addressing deficiencies in immune genes through therapeutic antibodies offers potential for patients with autosomal recessive or dominant deficiencies in vital genes like IRF-7,

IFNAR1, TLR-3. The identification of IFIT1, IFITM1, IRF7, ISG15, MX1, and OAS2 as pivotal components suggested their viability as potential drug targets for COVID-19 (*Prasad et al., 2020*). Similarly, targeting anti-type 1 IFN autoantibodies *via* antibody therapy can restore natural defenses, aiding controlling severe COVID-19. Counteracting inhibitory viral proteins (nsp16, nsp-1, NSP-8, NSP-9) through antibody design could restore effective type-1 IFN production by impeding their interaction with cellular components (*Beyer & Forero, 2022*). Engineering antibodies to enhance type-1 IFN production by targeting specific cellular factors presents a strategy for boosting antiviral immunity. Given the complexities, combining antibodies targeting distinct aspects of immune response, viral inhibition, and IFN production could offer a comprehensive treatment approach. Moreover, personalized antibody therapies tailored to individual genetic, immunological variations, and viral interactions could optimize specificity and efficacy in managing severe cases.

## MONOCLONAL ANTIBODIES AND INNATE IMMUNE TARGETS

Monoclonal antibodies (mAbs) have shown promising results for the effective management of COVID-19. These laboratory-engineered antibodies are designed to mimic the natural immune system response to infections. In the early COVID-19 pandemic, IgG mAbs directed against the spike protein of SARS-CoV-2, as single or in the form of mAb cocktails garnered substantial attention as a effective therapeutic solution for COVID-19 (*Focosi et al., 2022*). The Coronavirus Antibody Database (CoV-Ab Dab), managed by the University of Oxford, is a comprehensive repository containing 12,916 antibodies and nanobodies (as of the latest update on February 8, 2024) specifically designed to target SARS-CoV, MERS-CoV, and SARS-CoV-2 (*Raybould et al., 2021*). mAbs have also emerged as a promising class of therapeutics for targeting the innate immune response. These antibodies were developed with the goal of reducing the aberrant immune response observed in severe COVID-19 cases and target innate immune system components. These mAbs can block key pro-inflammatory cytokines such as IL-6 and TNF-α and several others, thereby weakening the cytokine storm associated with disease severity (*Abbasifard & Khorramdelazad, 2020*). Additionally, some mAbs directly target viral components, preventing the virus from evading innate immune detection and mounting a robust response (*Znaidia et al., 2022*). Several mAbs that may act upon the innate immune response are highlighted in Table 4. A recent study by *Sele et al. (2024)* highlight the pivotal role of SARS-CoV-2 non-structural protein 10 (nsp10) in enhancing the enzymatic activities of nsp14 and nsp16, crucial for the virus evasion of innate immunity. The research emphasizes the importance of the C-terminal region of nsp10 in its interaction with nsp14, highlighting the necessity of both N- and C-termini for optimal binding. Targeting these sites with mAbs could offer a promising strategy for combating SARS-CoV-2. The research and development of mAbs for COVID-19 should continue to evolve and explore as novel therapeutics. A deeper understanding of mAbs for fine-tuning the innate immune response can lead to the development of tailored therapeutics against SARS-CoV-2 and future variants.

**Table 4  Monoclonal antibodies as therapeutic agents against SARS-CoV-2, their targets, and outcomes.**

| S. No | Monoclonal antibody | Target | Outcomes | References |
|---|---|---|---|---|
| 1. | Tocilizumab, sarilumab | Antagonist to the IL-6 receptor | Reduce the cytokine storm | *Salama et al. (2021); Sciascia et al. (2020)* |
| 2. | Acalabrutinib, Ibrutinib, Acalabrutinib | Bruton's tyrosine kinase (BTK) inhibitor- impact on the signalling of TLRs, IL-1R, CD19, BCR, CXCR4, and Fcγ-R1 | Under clinical trial | *Roschewski et al. (2020)* |
| 3. | Anakinra, Gevokizumab, Canakinumab | Recombinant form antagonist of the IL-1 receptor | Minimizes hospitalization and death rate | *Cavalli et al. (2020), Freeman & Swartz (2020), Geranurimi et al. (2019), Huet et al. (2020), Zheng et al. (2019)* |
| 4. | Infliximab, adalimumab, certolizumab pegol | Anti-TNF-alpha antibody | Reduce the cytokine storm | *Keewan, Beg & Naser (2021), Valizadeh et al. (2020)* |
| 5. | Lenzilumab, Gimsilumab, Namilumab | Anti-GM-CSF monoclonal antibody | Reduce the cytokine storm | *Bonaventura et al. (2020)* |
| 6. | Sargramostim, Meptazinol | GM-CSF Partial opioid agonist | Reduce the cytokine storm | *Bonaventura et al. (2020), Lazarus & Gale (2021), Mihaescu et al. (2021)* |
| 7. | MSC-derived exosomes (MSC-Exo) | MSC-based therapy | Recruiting (under trail) | *Golchin, Seyedjafari & Ardeshirylajimi (2020), Mihaescu et al. (2021)* |
| 8. | Baricitinib | Inhibitor of Janus kinase (JAK) | Modulate the immune response and reduce inflammation | *Jorgensen et al. (2020), Kalil et al. (2021)* |
| 9. | Emapalumab | Targets interferon-gamma (IFN-gamma) | Reduce the cytokine storm | *Cure, Kucuk & Cure (2021)* |
| 10. | Clazakizumab | Targets IL-6 | Reduce the cytokine storm | *Lonze et al. (2022), Vaidya et al. (2020)* |
| 11. | Itolizumab | Targets CD6 | T-cell activation, modulates the immune response and reduces inflammation | *Díaz et al. (2020), Saavedra et al. (2020)* |
| 12. | Leronlimab (PRO160) | Target C-C chemokine receptor type 5 (CCR5). | Reduce the cytokine storm | *Agresti et al. (2021)* |
| 13. | Anifrolumab | Targets the type I interferon receptor | Modulating the immune response and in managing cytokine storm | *De Luca et al. (2020), Pourhoseingholi, Shojaee & Ashtari (2020)* |
| 14. | Mavrilimumab | Targets granulocyte-macrophage colony-stimulating factor receptor alpha (GM-CSFRα) | Anti-inflammatory effects, reduce the cytokine storm | *De Luca et al. (2020), Pourhoseingholi, Shojaee & Ashtari (2020)* |
| 15. | Camrelizumab | Targets programmed cell death protein 1 (PD-1) | Immunomodulatory effects | *AminJafari & Ghasemi (2020), Zhang et al. (2020c)* |

## Challenges of mAbs as therapeutics

SARS-CoV-2 undergoes rapid evolution and introduces certain mutations to its genome that may diminish the efficacy of antibody binding. Booster doses or novel formulations may be necessary in such cases. Production of monoclonal antibodies is a complex

process that requires costly biotechnological facilities thus elevated costs and reducing accessibility, particularly in low- and middle-income countries. Therapeutic antibodies exhibit optimal efficacy when administered at the initial stages of infection. In severe cases, hyperinflammatory responses such as cytokine storm may reduce its effectiveness. Most antibody therapies require intravenous injections within a healthcare facility, thereby constraining swift implementation of mAbs therapeutics. mAbs storage and transportation require strict cold-chain conditions, which introduce logistical challenges. mAbs treatment may pose risk of antibody-dependent enhancement (ADE) in rare instances and may facilitate viral infection instead of neutralizing it. Additionally, obtaining regulatory approval for novel antibodies requires significant time and resources. Researchers now a days are developing broad-spectrum antibodies, combination therapies, and long-acting monoclonal antibodies to improve efficacy and accessibility in response to the challenges posed.

## CONCLUSION AND PROSPECTS

The innate immune response is the host first line of defense for viral infection including SARS-CoV-2. A comprehensive study of this response in SARS-CoV-2 is crucial for developing effective strategies against SARS-CoV-2 variants. This knowledge will aid scientists and innovators in designing novel antibodies, enhancing preparedness for future outbreaks. Utilizing mAbs to rectify innate immune responses and counteract inhibitory viral proteins like nsp16, nsp-1, nsp-8, and nsp-9 offers innovative strategies to bolster anti-SARS-CoV-2 immunity and can be neutralized *via* the strategic design of antibodies that disrupt their interactions. Targeting anti-type 1 interferon (IFN) autoantibodies in severe cases allows for the restoration of the body natural defences, providing a tailored approach to control viral replication. The strategic design of antibodies not only neutralizes the inhibitory impact of viral proteins but also opens avenues for engineering antibodies capable of enhancing type-1 IFN production. This dual functionality contributes to a robust antiviral defence's mechanism. Considering the intricacies involved, a comprehensive strategy involving combination therapies, tailored to individual profiles, emerges as a promising frontier for treatment precision. Additionally, rectifying immune gene deficiencies such as IRF-7, IFNAR1, and TLR-3 can be accomplished by employing therapeutic antibodies as substitutes. Moreover, the integration of these approaches with the unique recognition capability of RIG-1, triggering interferon production in response to specific viral RNA structures, holds considerable promise for advancing targeted and efficient antibody-based therapies against COVID-19 disease. Understanding innate immune response and targeted therapy in the form of mAbs will be instrumental in addressing future outbreaks.

## ACKNOWLEDGEMENTS

The authors would like to thank Shuiab Altaf for his valuable contribution to pictorial representation of the review. We also acknowledge ChatGPT4 for language enhancement and improving flow of the manuscript.

### Funding

This work was supported by the Deanship of Scientific Research, Vice Presidency for Graduate Studies and Scientific Research, King Faisal University, Saudi Arabia (Grant No. KFU251646). The funders had no role in study design, data collection and analysis, decision to publish, or preparation of the manuscript.

### Grant Disclosures
The following grant information was disclosed by the authors:
Deanship of Scientific Research, Vice Presidency for Graduate Studies and Scientific Research, King Faisal University, Saudi Arabia: No. KFU251646.

### Competing Interests
The authors declare there are no competing interests.

### Author Contributions
- Mubashir Nazir conceived and designed the experiments, performed the experiments, analyzed the data, prepared figures and/or tables, authored or reviewed drafts of the article, and approved the final draft.
- Ishfaq Rashid Mir conceived and designed the experiments, performed the experiments, analyzed the data, prepared figures and/or tables, authored or reviewed drafts of the article, and approved the final draft.
- Shabir Ahmad Lone performed the experiments, analyzed the data, prepared figures and/or tables, authored or reviewed drafts of the article, and approved the final draft.
- Ghazala Muteeb performed the experiments, analyzed the data, prepared figures and/or tables, authored or reviewed drafts of the article, and approved the final draft.
- Ragib Alam performed the experiments, analyzed the data, prepared figures and/or tables, authored or reviewed drafts of the article, and approved the final draft.
- Anis Bashir Fomda performed the experiments, analyzed the data, prepared figures and/or tables, authored or reviewed drafts of the article, and approved the final draft.
- Nida Khan performed the experiments, analyzed the data, prepared figures and/or tables, authored or reviewed drafts of the article, and approved the final draft.
- Asim Azhar performed the experiments, analyzed the data, prepared figures and/or tables, authored or reviewed drafts of the article, and approved the final draft.
- Bashir Ahmad Fomda performed the experiments, analyzed the data, prepared figures and/or tables, authored or reviewed drafts of the article, and approved the final draft.
- Wajihul Hasan Khan conceived and designed the experiments, performed the experiments, analyzed the data, prepared figures and/or tables, authored or reviewed drafts of the article, and approved the final draft.

### Data Availability
This is a literature review.

Nazir et al. (2025), *PeerJ*, DOI 10.7717/peerj.19462

## Supplemental Information

Supplemental information for this article can be found online at http://dx.doi.org/10.7717/peerj.19462#supplemental-information.

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
