# Peer review of "Innate immunity, therapeutic targets and monoclonal antibodies in SARS-CoV-2 infection"

_PeerJ, doi:10.7717/peerj.19462_

## Round 0.1 · original submission · Major Revisions

Thank you for your manuscript. While the reviewers acknowledged the manuscript’s relevance and potential contribution to the field, several critical issues were identified, particularly regarding the structure, clarity, and depth of discussion. The reviewers highlighted areas needing improvement in terms of coherence, referencing, and the inclusion of updated and specific data. Please address the concerns raised by the reviewers and myself that are exposed next. It its important you address all of them to make sure the manuscript is sufficiently improved to be suitable for publication.

I found the introduction too broad and general, lacking a proper focus on immunity against SARS-CoV-2. Please, include specific details, such as the role of innate immunity and recent information on SARS-CoV-2 variants. Topics to add include differences in COVID-19 severity between immunocompromised and immunocompetent patients; severe COVID-19 association with inflammation and immune activation, and relevance of immunomodulatory therapies (e.g., corticosteroids, monoclonal antibodies).

Regarding abbreviations, make sure all of them (e.g., MDA5, LGP2, ISGs) are clear and consistently defined upon their first use to enhance clarity for a broader audience.

While you describe your review as systematic, the methodology for article selection remains unclear. Specify how you filtered articles from the vast number of PubMed results for “COVID-19 immunity”. Also, provide the total number of articles reviewed and justify their inclusion in the manuscript.

Several claims are unsupported by evidence or reference outdated sources. For instance, the statement about elevated IL-4 levels (lines 331-333) is based on studies lacking robust original data. Revise these claims and include recent and relevant references.

When discussing therapeutic strategies, critically assess their limitations and challenges.

Regarding the organization of the manuscript, I agree with the reviewers that it contains repetitions and disorganized sections (e.g., Toll-Like Receptors are discussed in two non-consecutive parts). To improve readability please use clear headings and subheadings to better organize discussions and shorten overly long paragraphs.

Please address reviewers' comments on figures and tables. Revise Figure 1 to ensure all elements are relevant to the manuscript’s focus. Add a detailed legend to Figure 2 explaining the molecular pathways presented. Include a diagram summarizing cGAS-STING signaling and therapeutic targets. Consider adding a column in Table 3 with information about protein structure/size.

Please also address the specifics that reviewers raised like providing more detail about chemokines such as CCL-2, CCL-3, and CCL-5; also include a separate figure summarizing apoptotic cell death and signaling during SARS-CoV-2 infection; and highlight recent therapeutic advances targeting the cGAS-STING pathway.

Thee conclusion should address challenges posed by new SARS-CoV-2 variants and critically evaluate therapeutic interventions discussed in the manuscript.

Besides all the comments above please attend the following key comments from each reviewer in your revision.

Reviewer 1

Address coherence issues and avoid irrelevant details (e.g., TLR7’s ability to identify synthetic oligoribonucleotides). Replace outdated references with more recent primary literature (e.g., Walls et al., Cell 2020). Revise the introduction and conclusion to provide a clearer focus.

Reviewer 2

Incorporate references suggested by the reviewer, if deemed relevant, such as studies on vitamin D receptor polymorphisms and thrombophilia genes in COVID-19 patients. Please do not feel obliged to include it if it is not appropriate.
Ensure references in tables are numbered consistently with the reference list.

Reviewer 3

Include cGAS-STING as a key therapeutic target in the abstract. Revise large paragraphs for better readability and include clear subheadings. Correct spelling errors (e.g., line 282).


Please submit a revised version of the manuscript that addresses all the points outlined above. Include a detailed response letter explaining how each reviewer comment was addressed or providing justification for any suggestions that were not implemented. Ensure that your revisions improve the manuscript’s scientific rigor, clarity, and overall quality.

We look forward to receiving your revised manuscript. Thank you!

Reviewer 1 ·

Basic reporting

The use of English is in most instances clear and correct. However, at some points, sentences are inadequate. There is some use of general, less academic words or phrases, while other parts are overly formal.

The main text is in parts incoherent, and the authors did not seem to differentiate relevant information from irrelevant. For instance, the text about Toll Like Receptors (lines 151-160) states that genes of TLR7 and TLR8 are capable of identifying synthetic oligoribonucleotides such as imidazoquinoline (irrelevant, in addition to a mistake in the sentence), while it misses the point that TLR7 might be interesting because of clinical differences in COVID_19 severity between male and female patients.

The structure of the text is unclear. For instance, Toll Like Receptors are addressed in 146-160 and later in 189-207.

Certain phrases are left without reference. For example, when the authors state that children exhibit higher basal expression of RIG-1 and MDA5 (lines 217-220).

The same sentence (lines 217-220) is irrelevant without mentioning that children generally have less severe COVID-19.

Literature references are provided. However, the authors often refer to review articles instead of original articles. For instance, in lines 79-80, the authors refer to two articles from 2004 and 2022, while a more appropriate reference would be from 2020, eg,: Walls et al, Cell 181, 281.292.e6

In multiple instances, the authors refer to expert opinions from 2020 when mentioning potential therapeutic agents, but many of these are outdated. These opinions are often outdated or not supported by evidence (eg. Prasad et al 2020).

The figures look mostly professional, apart from a detail in figure 1 (mAB:47D11 - unclear why this is showed).

In all tables, the authors refer to articles by numbers, but the reference list is not numbered. Therefore, it is impossible to check the adequacy.

The introduction is long, and too general, while lacking proper introduction of the subject (immunity against SARS-CoV-2). Therefore, the introduction does not match with the audience as stated in the article.

I would suggest revising the introduction completely. It can be helpful to include the following items:
o Immunocompromised patients are at risk for severe COVID-19
o Severe COVID-19 is associated with inflammation and immune activation
o Immunomodulation (corticosteroids, mAbs against certain interleukins, etc) can be helpful in preventing death
o Why does this review focus on innate immunity

Experimental design

The article content is within the aims and scope of the article.

In the ‘Search methodology’ section (lines 122-131), the authors suggest that this is a systematic research, with a reproducible search. However, the term ‘COVID-19 immunity’ has already 80,656 hits on Pubmed (on 2nd of September 2024), so how did the authors further select the articles that are reviewed here? In fact, the article seems a narrative review, without a reproducible search as such.

Validity of the findings

Most parts are written on a very general level, while the part on cGAS STING signalling pathway (lines 419-554) is disproportionately long and more detailed.

While the paper discusses the potential of monoclonal antibodies, it fails to critically assess their limitations and challenges.

The text contains many clainms that are not supported by evidence. For instance, the authors state in lines 331-333 that SARS-CoV-2 have increased levels of IL-4, but they refer to an article in which SARS-CoV-2 infection is not associated with increases of IL-4. The other article referred to claims that IL-4 is elevated in SARS-CoV-2 patients, but this is not supported by original data (Lu et al. Ebiomedicine 2020;55;102763).

A stronger critical evaluation would involve not just reporting what studies have found, but also discussing the quality of the evidence, potential biases, and the implications of these findings for clinical practice and future research.

In the conclusion, the authors state that the exploration of therapeutics for SARS-CoV-2 unveils a multifaceted approach, and this is followed by a number of hypothetical interventions that are not adequately addressed as such in the main text. In the last part, lines 649-654, the importance of non-structural protein 10 is introduced, but it remains unclear why, and why here. The conclusion also fails to address potential future challenges in terms of new viral variants. Therefore, I suggest to revise the conclusion completely.

Additional comments

This is a challenging, broad subject, and the scientific community can certainly benefit from a review on it. In parts, the manuscript adequately summarizes the existent literature. However, in other parts the manuscript contains incoherent sentences with irrelevant details or even mistakes.

·

Basic reporting

Clear and unambiguous, professional English used throughout.

Experimental design

Rigorous investigation performed to a high technical & ethical standard.

Validity of the findings

Impact and novelty not assessed. Meaningful replication encouraged where rationale & benefit to literature is clearly stated.

Additional comments

I kindly request that the necessary references be made in the introduction and discussion sections of the works that I am honored to participate in the work below.


Aci, Recai, et al. "Effect of vitamin D receptor gene BsmI polymorphism on hospitalization of SARS-CoV-2 positive patients." Nucleosides, Nucleotides & Nucleic Acids 43.3 (2024): 264-275.
Yigit, Serbulent, et al. "Vascular endothelial growth factor gene insertion/deletion polymorphism is associated with Vitamin D level in Turkish patients with coronavirus disease 2019." Revista da Associação Médica Brasileira 69.7 (2023).
Bilgin, Melek, et al. "Could SARS-CoV-2 Trigger the Formation of Antinuclear Antibodies?." Turkish Journal of Immunology 10.3 (2022).
Sezer, Ozlem, et al. "Possible effect of genetic background in thrombophilia genes on clinical severity of patients with coronavirus disease-2019: A prospective cohort study." Baghdad Journal of Biochemistry and Applied Biological Sciences 3.03 (2022): 183-199.

·

Basic reporting

The manuscript entitled "Innate immunity, therapeutic targets and monoclonal antibodies in SARS-CoV-2 infection" is a review highlighting the role of innate immunity against SARS-CoV-2 and summarising the possible treatment strategies targetting immunity. Overall, the manuscript is well written and tries to include recent findings on COVID-19 disease. However, I recommend some improvement.
1. The abstract section should include cGAS-STING signalling targets as important therapeutic strategies.
2. Introduction section should include information about recent COVID-19 variants.
3. Throughout the manuscript, there are numerous abbreviations used. All should be elaborated at the first appearance in the manuscript. For example, MDA5, LGP2 , and ISGs should be elaborated.
4. Adopttic cell death and signalling during SARS-CoV-2 can be summarized separately in a figure.
5. Need spelling correction at line 282
6. Other chemokines like CCL-2, CCL-3, CCL-5 should be explained in details as a chemokine
7. The cGAS-STING signalling pathway should be presented as a diagram, and recently discovered therapeutic strategies through this pathway should also included in the figure.
8. Figure 2 legend should contain the explanation of molecular pathway.
9. Another column with information about protein structure/ size can be added in Table 3

Experimental design

How many of the articles finally considered for this study Should be included in the method sections.
The writing style should include a heading and subheading for similar discussions. Some paragraphs are too much larger, and should be shortened for easy understanding and readability.

Validity of the findings

Valid

---

## Round 0.2 · Minor Revisions

I agree with the reviewers that the manuscript has improved substantially. However, one reviewer highlights remaining typographical issues (e.g., inconsistent paragraphing at lines 481 and 573, and a missing capital letter at line 575), and recommends a thorough proofreading. Please also follow their suggestion to break up overly long paragraphs to enhance readability. Kindly revise the manuscript accordingly and provide a response to the reviewer’s comments. Once these minor issues are addressed, I will be able to recommend the manuscript for acceptance. Thank you

·

Basic reporting

I don't have anything to add

Experimental design

I don't have anything to add

Validity of the findings

I don't have anything to add

Additional comments

I don't have anything to add

·

Basic reporting

The manuscript entitled "Innate Immunity, Therapeutic Targets and Monoclonal Antibodies in SARS-CoV-2 Infection" has been well improved after revision; just some typographical issues need to be addressed (e.g., inconsistent paragraphing at lines 481 and 573). Capital letter is necessary at 575. Such problems need to be addressed throughout the manuscript. Still, some paragraphs are too long and need to be shortened or broken down into multiple paragraphs.

Experimental design

Well designed.

Validity of the findings

Findings are well applicable for researchers and clinicians in the related fields.

Additional comments

The manuscript improved well after revision, and the authors tried to address most of the comments from the editor and reviewers.

---

## Round 0.3 · accepted · Accept

The evaluation of your revision is completed. The last round of review involved only minor points, and the reviewer has since recommended acceptance.
I am satisfied that all the concerns raised during the peer review process have been resolved, and I believe the manuscript meets the standards expected by PeerJ.

I am pleased to inform you that your manuscript has been accepted for publication. Congratulations!

·

Basic reporting

The current version is well-improved and acceptable for publication.

Experimental design

Well designed

Validity of the findings

Findings are interesting and might be useful for the scientific community.